# Rapid Neural Architecture Search by Learning to Generate Graphs from Datasets

**Hayeon Lee**[1]*,  **Eunyoung Hyung**[1]*,   **Sung Ju Hwang**[1,2]
KAIST[1], AITRICS[2], South Korea
{hayeon926, eunyoung0301, sjhwang82}@kaist.ac.kr

## Abstract

Despite the success of recent Neural Architecture Search (NAS) methods on various tasks which have shown to output networks that largely outperform human-designed networks, conventional NAS methods have mostly tackled the optimization of searching for the network architecture for a single task (dataset), which does not generalize well across multiple tasks (datasets). Moreover, since such task-specific methods search for a neural architecture from scratch for every given task, they incur a large computational cost, which is problematic when the time and monetary budget are limited. In this paper, we propose an efficient NAS framework that is trained once on a database consisting of datasets and pretrained networks and can rapidly search for a neural architecture for a novel dataset. The proposed MetaD2A (Meta Dataset-to-Architecture) model can stochastically generate graphs (architectures) from a given set (dataset) via a cross-modal latent space learned with amortized meta-learning. Moreover, we also propose a meta-performance predictor to estimate and select the best architecture without direct training on target datasets. The experimental results demonstrate that our model meta-learned on subsets of ImageNet-1K and architectures from NAS-Bench 201 search space successfully generalizes to multiple unseen datasets including CIFAR-10 and CIFAR-100, with an average search time of 33 GPU seconds. Even under MobileNetV3 search space, MetaD2A is 5.5K times faster than NSGANetV2, a transferable NAS method, with comparable performance. We believe that the MetaD2A proposes a new research direction for rapid NAS as well as ways to utilize the knowledge from rich databases of datasets and architectures accumulated over the past years. Code is available at https://github.com/HayeonLee/MetaD2A.

## 1 Introduction

The rapid progress in the design of neural architectures has largely contributed to the success of deep learning on many applications (Krizhevsky et al., 2012; Cho et al., 2014; He et al., 2016; Szegedy et al.; Vaswani et al., 2017; Zhang et al., 2018). However, due to the vast search space, designing a novel neural architecture requires a time-consuming trial-and-error search by human experts. To tackle such inefficiency in the manual architecture design process, researchers have proposed various *Neural Architecture Search (NAS)* methods that automatically search for optimal architectures, achieving models with impressive performances on various tasks that outperform human-designed counterparts (Baker et al., 2017; Zoph & Le, 2017; Kandasamy et al., 2018; Liu et al., 2018; Luo et al., 2018; Pham et al., 2018; Liu et al., 2019; Xu et al., 2020; Chen et al., 2021).

Recently, large benchmarks for NAS (NAS-101, NAS-201) (Ying et al., 2019; Dong & Yang, 2020) have been introduced, which provide databases of architectures and their performances on benchmark datasets. Yet, most conventional NAS methods cannot benefit from the availability of such databases, due to their task-specific nature which requires repeatedly training the model from scratch for each new dataset (See Figure 1 Left). Thus, searching for an architecture for a new task (dataset) may require a large number of computations, which may be problematic when the time and mon-

---

*These authors contributed equally to this work.

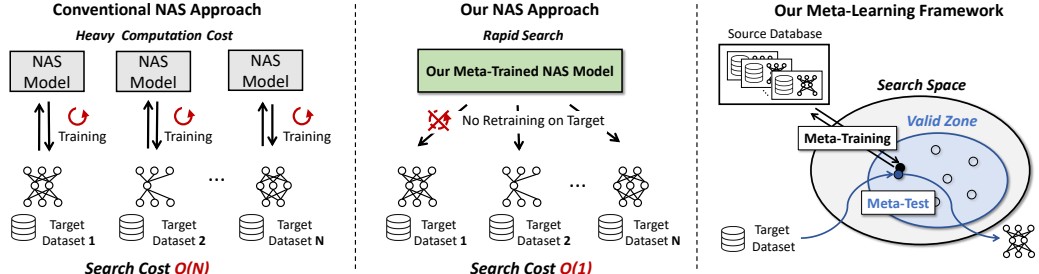

Figure 1: **Left:** Most conventional NAS approaches need to repeatedly train NAS model on each given target dataset, which results in enormous total search time on multiple datasets. **Middle:** We propose a novel NAS framework that generalizes to any new target dataset to generate specialized neural architecture without additional NAS model training after only meta-training on the source database. Thus, our approach cut down the search cost for training NAS model on multiple datasets from O(N) to O(1). **Right:** For unseen target dataset, we utilize amortized meta-knowledge represented as set-dependent architecture generative representations.

etary budget are limited. How can we then exploit the vast knowledge of neural architectures that have been already trained on a large number of datasets, to better generalize over an unseen task?

In this paper, we introduce amortized meta-learning for NAS, where the goal is to learn a NAS model that generalizes well over the task distribution, rather than a single task, to utilize the accumulated meta-knowledge to new target tasks. Specifically, we propose an efficient NAS framework that is trained once from a database containing datasets and their corresponding neural architectures and then generalizes to multiple datasets for searching neural architectures, by learning to generate a neural architecture from a given dataset. The proposed MetaD2A (Meta Dataset-to-Architecture) framework consists of a set encoder and a graph decoder, which are used to learn a cross-modal latent space for datasets and neural architectures via amortized inference. For a new dataset, MetaD2A stochastically generates neural architecture candidates from set-dependent latent representations, which are encoded from a new dataset, and selects the final neural architecture based on their predicted accuracies by a performance predictor, which is also trained with amortized meta-learning. The proposed meta-learning framework reduces the search cost from O(N) to O(1) for multiple datasets due to no training on target datasets. After one-time building cost, our model only takes just a few GPU seconds to search for neural architecture on an unseen dataset (See Figure 1).

We meta-learn the proposed MetaD2A on subsets of ImageNet-1K and neural architectures from the NAS-Bench201 search space. Then we validate it to search for neural architectures on multiple unseen datasets such as MNIST, SVHN, CIFAR-10, CIFAR-100, Aircraft, and Oxford-IIIT Pets. In this experiment, our meta-learned model obtains a neural architecture within 33 GPU seconds on average without direct training on a target dataset and largely outperforms all baseline NAS models. Further, we compare our model with representative transferable NAS method (Lu et al., 2020) on MobileNetV3 search space. We meta-learn our model on subsets of ImageNet-1K and neural architectures from the MobileNetV3 search space. The meta-learned our model successfully generalizes, achieving extremely fast search with competitive performance on four unseen datasets such as CIFAR-10, CIFAR-100, Aircraft, and Oxford-IIIT Pets.

To summarize, our contribution in this work is threefold:

- We propose a novel NAS framework, MetaD2A, which rapidly searches for a neural architecture on a new dataset, by sampling architectures from latent embeddings of the given dataset then selecting the best one based on their predicted performances.

- To this end, we propose to learn a cross-modal latent space of datasets and architectures, by performing amortized meta-learning, using a set encoder and a graph decoder on subsets of ImageNet-1K.

- The meta-learned our model successfully searches for neural architectures on multiple unseen datasets and achieves state-of-the-art performance on them in NAS-Bench201 search space, especially searching for architectures within 33 GPU seconds on average.

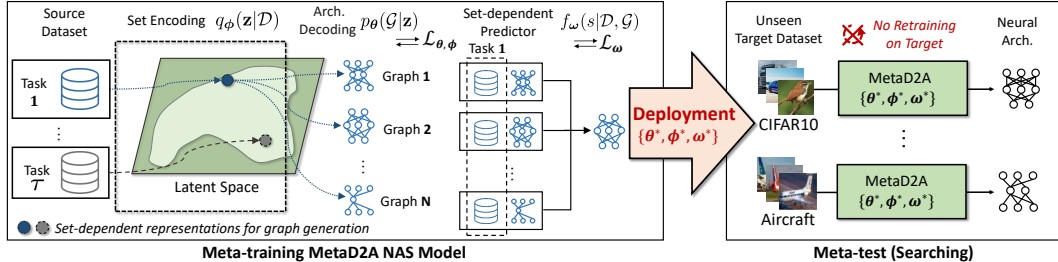

Figure 2: **Overview of MetaD2A** The proposed generator with $\theta$ and $\phi$ meta-learns *the set-dependent graph representations* on the meta-training tasks, where each task contains a subset of ImageNet-1K and high-quality architecture for the subset. The proposed predictor with $\omega$ meta-learns to predict performance, considering the dataset as well as the graph. In the meta-test (searching) phase, the meta-learned MetaD2A generalizes to output set-specialized neural architecture for new target datasets without additional NAS model training.

## 2 RELATED WORK

**Neural Architecture Search (NAS)** NAS is an automated architecture search process which aims to overcome the suboptimality of manual architecture designs when exploring the extensive search space. NAS methods can be roughly categorized into reinforcement learning-based methods (Zoph & Le, 2017; Zoph et al., 2018; Pham et al., 2018), evolutionary algorithm-based methods (Real et al., 2019; Lu et al., 2020), and gradient-based methods (Liu et al., 2019; Cai et al., 2019; Luo et al., 2018; Dong & Yang, 2019b; Chen et al., 2021; Xu et al., 2020; Fang et al., 2020). Among existing approaches, perhaps the most relevant approach to ours is NAO (Luo et al., 2018), which maps DAGs onto a continuous latent embedding space. However, while NAO performs graph reconstruction for a single task, ours generates data-dependent Directed Acyclic Graphs (DAGs) across multiple tasks. Another important open problem in NAS is reducing the tremendous computational cost resulting from the large search space (Cai et al., 2019; Liu et al., 2018; Pham et al., 2018; Liu et al., 2019; Chen et al., 2021). GDAS (Dong & Yang, 2019b) tackles this by optimizing sampled sub-graphs of DAG. PC-DARTS (Xu et al., 2020) reduces GPU overhead and search time by partially selecting channel connections. However, due to the task-specific nature of those methods, they should be retrained from the scratch for each new unseen task repeatedly and each will take a few GPU hours. The accuracy-predictor-based transferable NAS called NSGANetV2 (Lu et al., 2020) alleviates this issue by adapting the ImageNet-1K pre-trained network to multiple target datasets, however, this method is still expensive due to adapting procedure on each dataset.

**Meta-learning** Meta-learning (learning to learn) aims to train a model to generalize over a distribution of tasks, such that it can rapidly adapt to a new task (Vinyals et al., 2016; Snell et al., 2017; Finn et al., 2017; Nichol et al., 2018; Lee et al., 2019b; Hou et al., 2019). Recently, LEO (Rusu et al., 2019) proposed a scalable meta-learning framework which learns the latent generative representations of model parameters for a given data in a low-dimensional space for few-shot classification. Similarly to LEO (Rusu et al., 2019), our method learns a low-dimensional latent embedding space, but we learn a cross-modal space for both datasets and models for task-dependent model generation.

**Neural Architecture Search with Meta-Learning** Recent NAS methods with gradient-based meta-learning (Elsken et al., 2020; Lian et al., 2019; Shaw et al., 2019) have shown promising results on adapting to different tasks. However, they are only applicable on small scale tasks such as few-shot classification tasks (Elsken et al., 2020; Lian et al., 2019) and require high-computation time, due to the multiple unrolling gradient steps for one meta-update of each task. While some attempt to bypass the bottleneck with a first-order approximation (Lian et al., 2019; Shaw et al., 2019) or parallel computations with GPUs (Shaw et al., 2019), but their scalability is intrinsically limited due to gradient updates over a large number of tasks. To tackle such a scalability issue, we perform amortized inference over the multiple tasks by encoding a dataset into the low-dimensional latent vector and exploit fast GNN propagation instead of the expensive gradient update.

## 3 METHOD

Our goal is to output a high-performing neural architecture for a given dataset rapidly by learning the prior knowledge obtained from the rich database consisting of datasets and their corresponding

neural architectures. To this end, we propose Meta Dataset-to-Architecture (MetaD2A) framework which learns the cross-modal latent space of datasets and their neural architectures. Further, we introduce a meta-performance predictor, which predicts accuracies of given architectures without training the predictor on an unseen target dataset. Overview of the proposed approach is illustrated in Figure 1.

### 3.1 META-TRAINING NAS MODEL

To formally define the problem, let us assume that we have a source database of $N_\tau$ number of tasks, where each task $\tau = \{\mathcal{D}, \mathcal{G}, s\}$ consists of a dataset $\mathcal{D}$, a neural architecture represented as a Directed Acyclic Graph (DAG) $\mathcal{G}$ and an accuracy $s$ obtained from the neural architecture $\mathcal{G}$ trained on $\mathcal{D}$. In the meta-training phase, the both dataset-to-architecture generator and meta-predictor learn to generalize over task distribution $p(\tau)$ using the source database. We describe how to empirically construct source database in Section 4.1.1.

#### 3.1.1 LEARNING TO GENERATE GRAPHS FROM DATASETS

We propose a dataset-to-architecture generator which takes a dataset and then generates high-quality architecture candidates for the set. We want the generator to generate even novel architectures, which are not contained in the source database, at meta-test. Thus, the generator learns the *continuous* cross-modal latent space $\mathcal{Z}$ of datasets and neural architectures from the source database. For each task $\tau$, the generator encodes dataset $\mathcal{D}$ as a vector $\mathbf{z}$ through the set encoder $q_\phi(\mathbf{z}|\mathcal{D})$ parameterized by $\phi$ and then decodes a new graph $\tilde{\mathcal{G}}$ from $\mathbf{z}$ which are sampled from the prior $p(\mathbf{z})$ by using the graph decoder $p_\theta(\mathcal{G}|\mathbf{z})$ parameterized by $\theta$. Then, our goal is that $\tilde{\mathcal{G}}$ generated from $\mathcal{D}$ to be the true $\mathcal{G}$ which is pair of $\mathcal{D}$. We meta-learn the generator using set-amortized inference, by maximizing the approximated evidence lower bound (ELBO) as follows:

$$\max_{\phi, \theta} \sum_{\tau \sim p(\tau)} \mathcal{L}_{\phi, \theta}^\tau (\mathcal{D}, \mathcal{G}) \tag{1}$$

where

$$\mathcal{L}_{\phi, \theta}^\tau (\mathcal{D}, \mathcal{G}) = \mathbb{E}_{\mathbf{z} \sim q_\phi(\mathbf{z}|\mathcal{D})} \big[ \log p_\theta (\mathcal{G}|\mathbf{z}) \big] - \lambda \cdot L_{KL}^\tau \big[ q_\phi (\mathbf{z}|\mathcal{D}) || p(\mathbf{z}) \big] \tag{2}$$

Each dimension of the prior $p(\mathbf{z})$ factorizes into $\mathcal{N}(0, 1)$. $L_{KL}^\tau$ is the KL divergence between two multivariate Gaussian distributions which has a simple closed form (Kingma & Welling, 2014) and $\lambda$ is the scalar weighting value. Using the reparameterization trick on $\mathbf{z}$, we optimize the above objective by stochastic gradient variational Bayes (Kingma & Welling, 2014). We use a set encoder described in Section 3.1.3 and we adopt a Graph Neural Network (GNN)-based decoder for directed acyclic graph (DAG)s (Zhang et al., 2019), which allows message passing to happen only along the topological order of the DAGs. For detailed descriptions for the generator, see Section A of Suppl.

#### 3.1.2 META-PERFORMANCE PREDICTOR

While many performance predictor for NAS have been proposed (Luo et al., 2018; Cai et al., 2020; Lu et al., 2020; Zhou et al., 2020; Tang et al., 2020), those performance predictors repeatedly collect architecture-accuracy database for each new dataset, which results in huge total cost on many datasets. Thus, the proposed predictor $f_\omega(s|\mathcal{D}, \mathcal{G})$ takes a **dataset** as well as graph as an input to support multiple datasets, while the existing performance predictor takes a graph only. Then, the proposed predictor meta-learns set-dependent performance proxy generalized over the task distribution $p(\tau)$ in the meta-training stage. This allows the meta-learned predictor to accurately predict performance on unseen datasets without additional training. The proposed predictor $f_\omega$ consists of a dataset encoder and a graph encoder, followed by two linear layers with $relu$. For dataset encoding, we use the set encoder of Section 3.1.3 which takes $\mathcal{D}$ as an input. We adopt direct acyclic graph encoder (Zhang et al., 2019) for DAG $\mathcal{G}$ (Please refer to Section B of Suppl.). We concatenate the outputs of both graph encoder and the set encoder, and feed them to two linear layers with $relu$ to predict accuracy. We train the predictor $f_\omega$ to minimize the MSE loss $\mathcal{L}_\omega^\tau(s, \mathcal{D}, \mathcal{G})$ between the predicted accuracy and the true accuracy $s$ of the model on each task sampled from the source database:

$$\min_\omega \sum_{\tau \sim p(\tau)} \mathcal{L}_\omega^\tau(s, \mathcal{D}, \mathcal{G}) = \sum_{\tau \sim p(\tau)} (s - f_\omega(\mathcal{D}, \mathcal{G}))^2 \tag{3}$$

### 3.1.3 Set Encoder

The efficacy of the proposed framework is dependent on how accurately set encoder captures the distribution of the target dataset and extracts information related with the goal of the generator and the predictor. To compress the entire instances from a dataset $\mathcal{D}$ into a single latent code $\mathbf{z}$, the set encoder should process input sets of any size and summarize consistent information agnostically to the order of the instances (permutation-invariance). Existing set encoders such as *DeepSet* (Zaheer et al., 2017), *SetTransformer* (Lee et al., 2019a), and *StatisticsPooling* (Lee et al., 2020) fulfill those requirements and might be used. However, *DeepSet* and *SetTransformer* are non-hierarchical poolings, thus cannot accurately model individual classes in the given dataset. Moreover, *DeepSet* and *StatisticsPooling* resort to simple averaging of the instance-wise representations.

Therefore, we introduce a novel set encoder which stacks two permutation-invariant modules with attention-based learnable parameters. The lower-level intra-class encoder captures the class prototypes that reflect label information, and the high-level inter-class encoder considers the relationship between class prototypes and aggregates them into a latent vector. The proposed structure of the set encoder models high-order interactions between the set elements allowing the generator and predictor to effectively extract useful information to achieve each goal.

Specifically, for a given dataset $\mathcal{D} = \{\boldsymbol{X}, \boldsymbol{Y}\}$, where $\boldsymbol{X} = \{\boldsymbol{X}_c\}_{c=1}^C$ and $\boldsymbol{Y} = \{\boldsymbol{Y}_c\}_{c=1}^C$ are the set of instances and target labels of $C$ classes respectively. We randomly sample instances $\{\mathbf{x}|\mathbf{x} \in \boldsymbol{B}_c\} \in \mathbb{R}^{b_c \times d_x}$ of class $c$, where $\mathbf{x}$ is a $d_x$ dimensional feature vector, $\boldsymbol{B}_c \subset \boldsymbol{X}_c$ and $||\boldsymbol{B}_c|| = b_c$. We input the sampled instances into the IntraSetPool, the intra-class encoder, to encode class prototype $\mathbf{v}_c \in \mathbb{R}^{1 \times d_{v_c}}$ for each class $c = 1, ..., C$. Then we further feed the class-specific set representations $\{\mathbf{v}_c\}_{c=1}^C$ into the InterSetPool, the inter-class encoder, to generate the dataset representation $\mathbf{h}_e \in \mathbb{R}^{1 \times d_{h_e}}$ as follows:

$$\mathbf{v}_c = \text{IntraSetPool}\big(\{\mathbf{x}|\mathbf{x} \in \boldsymbol{B}_c\}\big), \quad \mathbf{h}_e = \text{InterSetPool}\big(\{\mathbf{v}_c\}_{c=1}^C\big) \tag{4}$$

Both the set poolings are stacked attention-based blocks borrowed from Lee et al. (2019a). Note that while Lee et al. (2019a) is an attention-based set encoder, it ignores class label information of given dataset, which may lead to poor performance. Please see Section C of the Suppl. for more details.

## 3.2 Meta-Test (Searching)

In the meta-test stage, for an unseen dataset $\hat{\mathcal{D}}$, we can obtain $n$ set-dependent DAGs $\{\hat{\mathcal{G}}_i\}_{i=1}^n$, with the meta-trained generator parameterized by $\phi^*$ and $\boldsymbol{\theta}^*$, by feeding $\hat{\mathcal{D}}$ as an input. Through such set-level amortized inference, our method can easily generate neural architecture(s) for the novel dataset. The latent code $\mathbf{z} \in \mathbb{R}^{1 \times d_z}$ can be sampled from a dataset-conditioned Gaussian distribution with diagonal covariance where $\text{NN}_{\boldsymbol{\mu}}, \text{NN}_{\boldsymbol{\sigma}}$ are single linear layers:

$$\mathbf{z} \sim q_\phi(\mathbf{z}|\mathcal{D}) = \mathcal{N}(\boldsymbol{\mu}, \boldsymbol{\sigma}^2) \quad \text{where} \quad \boldsymbol{\mu}, \boldsymbol{\sigma} = \text{NN}_{\boldsymbol{\mu}}(\mathbf{h}_e), \text{NN}_{\boldsymbol{\sigma}}(\mathbf{h}_e) \tag{5}$$

In the meta-test, the predictor $f_{\boldsymbol{\omega}^*}(\hat{s}_i|\hat{\mathcal{D}}, \hat{\mathcal{G}}_i)$ predicts accuracies $\{\hat{s}_i\}_{i=1}^n$ for a given unseen dataset $\hat{\mathcal{D}}$ and each generated architecture of $\{\hat{\mathcal{G}}_i\}_{i=1}^n$ and then select the neural architecture having the highest predicted accuracy among $\{\hat{s}_i\}_{i=1}^n$.

## 4 Experiment

We conduct extensive experiments to validate MetaD2A framework. First, we compare our model with conventional NAS methods on NAS-Bench-201 search space in Section 4.1. Second, we compare our model with transferable NAS method under a large search space in Section 4.2. Third, we compare our model with other Meta-NAS approaches on few-shot classification tasks in Section 4.3. Finally, we analyze the effectiveness of our framework in Section 4.4.

### 4.1 NAS-Bench-201 Search Space

#### 4.1.1 Experiment Setup

We learn our model on source database consisting of subsets of ImageNet-1K and neural architectures of NAS-Bench-201 (Dong & Yang, 2020) and (meta-)test our model by searching for architectures on 6 benchmark datasets without additional NAS model training.

**NAS-Bench-201 search space** contains cell-based neural architectures, where each cell is represented as directed ayclic graph (DAG) consisting of the 4 nodes and the 6 edge connections. For each edge connection, NAS models select one of 5 operation candidates such as zerorize, skip connection, 1-by-1 convolution, 3-by-3 convolution, and 3-by-3 average pooling.

**Source Database** To meta-learn our model, we practically collect multiple tasks where each task consists of (dataset, architecture, accuracy). We compile ImageNet-1K (Deng et al., 2009) as multiple sub-sets by randomly sampling 20 classes with an average of 26K images for each sub-sets and assign them to each task. All images are downsampled by $32 \times 32$ size. We search for the set-specific architecture of each sampled dataset using random search among high-quality architectures which are included top-5000 performance architecture group on ImageNet-16-120 or GDAS (Dong & Yang, 2019b). For the predictor, we additionally collect 2,920 tasks through random sampling. We obtain its accuracy by training the architecture on dataset of each task. We collect $N_\tau =$1,310/4,230 meta-training tasks for the generator/predictor and 400/400 meta-validation tasks for them, respectively. Meta-training time is 12.7/8.4 GPU hours for the generator/the predictor and note that meta-training phase is needed only once for all experiments of NAS-Bench-201 search space.

**Meta-Test Datasets** We apply our model trained from source database to 6 benchmark datasets such as 1) CIFAR-10 (Krizhevsky et al., 2009), 2) CIFAR-100 (Krizhevsky et al., 2009), 3) MNIST (Le-Cun & Cortes, 2010), 4) SVHN (Netzer et al., 2011), 5) Aircraft (Maji et al., 2013), and 6) Oxford-IIIT Pets (Parkhi et al., 2012). On CIFAR10 and CIFAR100, the generator generates 500 neural architectures and we select 30 architectures based on accuracies predicted by the predictor. Following SETN (Dong & Yang, 2019a), we retrieve the accuracies of $N$ architecture candidates from the NAS-bench-201 and report the highest final accuracy for each run. While $N = 1000$ in SETN, we set a smaller number of samples ($N = 30$) for MetaD2A. We report the mean accuracies over 10 runs of the search process by retrieving accuracies of searched architectures from NAS-Bench-201. On MNIST, SVHN, Aircraft, and Oxford-IIIT Pets, the generator generates 50 architectures and select the best one with the highest predicted accuracy. we report the accuracy averaging over 3 runs with different seeds. For fair comparison, the searched architectures from our model are trained on each target datasets from the scratch. Note that once trained MetaD2A can be used for more datasets without additional training. Our model is performed with a single Nvidia 2080ti GPU.

### 4.1.2 RESULTS ON UNSEEN DATASETS

Table 1 shows that our model meta-learned on the source database can successfully generalize to 6 unseen datasets such as MNIST, SVHN, CIFAR-10, CIFAR-100, Aircraft, and Oxford-IIIT Pets by outperforming all baselines. Since the meta-learned MetaD2A can output set-specialized architectures on target datasets through inference process with **no** training cost, the search speed is extremely fast. As shown in Table 1, the search time of MetaD2A averaging on 6 benchmark datasets is within 33 GPU second. This is impressive results in that it is at least $147\times$ (maximum: $12169\times$) faster than conventional set-specific NAS approaches which need training NAS models on each target dataset. Such rapid search of REA, RS, REINFORCE and BOHB is only possible where all of the accuracies are pre-computed like NAS-Bench201 so that it can retrieve instantly on the target dataset, therefore, it is difficult to apply them to other non-benchmark datasets. Especially, we observe that MetaD2A which is learned over multiple tasks benefit to search set-dependent neural architectures for fine-grained datasets such as Aircraft and Oxford-IIIT Pets.

## 4.2 MOBILENETV3 SEARCH SPACE

### 4.2.1 EXPERIMENT SETUP

We apply our meta-trained model on four unseen datasets, comparing with transferable NAS (NS-GANetV2 (Lu et al., 2020)) under the same search space of MobileNetV3, where it contains more than $10^{19}$ architectures. Each CNN architecture consists of five sequential blocks and the targets of searching are the number of layers, the number of channels, kernel size, and input resolutions. For a fair comparison, we also exploit the supernet for the parameters as NSGANetV2 does. We collect $N_\tau = 3,018/153,408$ meta-training tasks for the generator/predictor and $646/32,872$ meta-validation tasks, respectively as a source database from the ImageNet-1K dataset and architectures of MobileNetV3 search space. Meta-training time is 2.21/1.41 GPU days for the generator/the predictor. Note that the meta-training phase is needed only **once** on the source database.

Table 1: **Performance on Unseen Datasets (Meta-Test)** MetaD2A conducts amortized inference on unseen target datasets after meta-training on source database consisting of subsets of ImageNet-1K and architectures of NAS-Bench-201 search space. Meta-training time is 12.7/8.4 GPU hours for the generator/the predictor. For fair comparison, the parameters of searched architectures are trained on each dataset from scratch instead of transferring parameters from ImageNet. $T$ is the time to construct precomputed architecture database for each target. We report accuracies with 95% confidence intervals.

| Target Dataset | NAS Method | NAS Training-free on Target | Params (M) | Search Time (GPU Sec) | Speed Up | Search Cost ($) | Accuracy (%) |
|---|---|---|---|---|---|---|---|
| CIFAR-10 | ResNet (He et al., 2016) | | 0.86 | N/A | N/A | N/A | $93.97_{\pm0.00}$ |
| | REA (Real et al., 2019) | | - | 0.02+$T$ | - | - | $93.92_{\pm0.30}$ |
| | RS (Bergstra & Bengio, 2012) | | - | 0.01+$T$ | - | - | $93.70_{\pm0.36}$ |
| | REINFORCE (Williams, 1992) | | - | 0.12+$T$ | - | - | $93.85_{\pm0.37}$ |
| | BOHB (Falkner et al., 2018) | | - | 3.59+$T$ | - | - | $93.61_{\pm0.52}$ |
| | RSPS (Li & Talwalkar, 2019) | | - | 10200 | 147× | 4.13 | $84.07_{\pm3.61}$ |
| | SETN (Dong & Yang, 2019a) | | - | 30200 | 437× | 12.25 | $87.64_{\pm0.00}$ |
| | GDAS (Dong & Yang, 2019b) | | - | 25077 | 363× | 10.17 | $93.61_{\pm0.09}$ |
| | PC-DARTS (Xu et al., 2020) | | 1.17 | 10395 | 150× | 4.21 | $93.66_{\pm0.17}$ |
| | DrNAS (Chen et al., 2021) | | 1.53 | 21760 | 315× | 8.82 | $94.36_{\pm0.00}$ |
| | **MetaD2A (Ours)** | ✓ | 1.11 | **69** | 1× | **0.028** | $\mathbf{94.37_{\pm0.03}}$ |
| CIFAR-100 | ResNet (He et al., 2016) | | 0.86 | N/A | N/A | N/A | $70.86_{\pm0.00}$ |
| | REA (Real et al., 2019) | | - | 0.02+T | - | - | $71.84_{\pm0.99}$ |
| | RS (Bergstra & Bengio, 2012) | | - | 0.01+T | - | - | $71.04_{\pm1.07}$ |
| | REINFORCE (Williams, 1992) | | - | 0.12+T | - | - | $71.71_{\pm1.09}$ |
| | BOHB (Falkner et al., 2018) | | - | 3.59+T | - | - | $70.85_{\pm1.28}$ |
| | RSPS (Li & Talwalkar, 2019) | | - | 18841 | 196× | 7.64 | $52.31_{\pm5.77}$ |
| | SETN (Dong & Yang, 2019a) | | - | 58808 | 612× | 23.85 | $59.09_{\pm0.24}$ |
| | GDAS (Dong & Yang, 2019b) | | - | 51580 | 537× | 20.91 | $70.70_{\pm0.30}$ |
| | PC-DARTS (Xu et al., 2020) | | 0.26 | 19951 | 207× | 8.09 | $66.64_{\pm2.34}$ |
| | DrNAS (Chen et al., 2021) | | 1.20 | 34529 | 359× | 14.00 | $\mathbf{73.51_{\pm0.00}}$ |
| | **MetaD2A (Ours)** | ✓ | 1.07 | **96** | 1× | **0.039** | $\mathbf{73.51_{\pm0.00}}$ |
| MNIST | ResNet (He et al., 2016) | | 0.86 | N/A | N/A | N/A | $99.67_{\pm0.01}$ |
| | RSPS (Li & Talwalkar, 2019) | | 0.25 | 22457 | 3208× | 9.10 | $99.63_{\pm0.02}$ |
| | SETN (Dong & Yang, 2019a) | | 0.56 | 69656 | 9950× | 28.24 | $99.69_{\pm0.04}$ |
| | GDAS (Dong & Yang, 2019b) | | 0.82 | 60186 | 8598× | 24.40 | $99.64_{\pm0.04}$ |
| | PC-DARTS (Xu et al., 2020) | | 0.62 | 24857 | 3551× | 10.08 | $99.66_{\pm0.04}$ |
| | DrNAS (Chen et al., 2021) | | 1.53 | 44131 | 6304× | 17.89 | $99.59_{\pm0.02}$ |
| | **MetaD2A (Ours)** | ✓ | 0.61 | **7** | 1× | **0.002** | $\mathbf{99.71_{\pm0.08}}$ |
| SVHN | ResNet (He et al., 2016) | | 0.86 | N/A | N/A | N/A | $96.13_{\pm0.19}$ |
| | RSPS (Li & Talwalkar, 2019) | | 0.48 | 27962 | 3994× | 11.34 | $96.17_{\pm0.12}$ |
| | SETN (Dong & Yang, 2019a) | | 0.48 | 85189 | 12169× | 34.54 | $96.02_{\pm0.12}$ |
| | GDAS (Dong & Yang, 2019b) | | 0.24 | 71595 | 10227× | 10.17 | $95.57_{\pm0.57}$ |
| | PC-DARTS (Xu et al., 2020) | | 0.47 | 31124 | 4446× | 12.62 | $95.40_{\pm0.67}$ |
| | DrNAS (Chen et al., 2021) | | 1.53 | 52791 | 7541× | 21.40 | $96.30_{\pm0.05}$ |
| | **MetaD2A (Ours)** | ✓ | 0.86 | **7** | 1× | **0.004** | $\mathbf{96.34_{\pm0.37}}$ |
| Aircraft | ResNet (He et al., 2016) | | 0.86 | N/A | N/A | N/A | $47.01_{\pm1.16}$ |
| | RSPS (Li & Talwalkar, 2019) | | 0.22 | 18697 | 1869× | 7.58 | $42.19_{\pm3.88}$ |
| | SETN (Dong & Yang, 2019a) | | 0.44 | 18564 | 1856× | 7.52 | $44.84_{\pm3.96}$ |
| | GDAS (Dong & Yang, 2019b) | | 0.62 | 18508 | 1850× | 7.50 | $53.52_{\pm0.48}$ |
| | PC-DARTS (Xu et al., 2020) | | 0.32 | 3524 | 352× | 1.42 | $26.33_{\pm3.40}$ |
| | DrNAS (Chen et al., 2021) | | 1.03 | 34529 | 3452× | 13.14 | $46.08_{\pm7.00}$ |
| | **MetaD2A (Ours)** | ✓ | 0.83 | **10** | 1× | **0.004** | $\mathbf{58.43_{\pm1.18}}$ |
| Oxford-IIIT Pets | ResNet (He et al., 2016) | | 0.86 | N/A | N/A | N/A | $25.58_{\pm3.43}$ |
| | RSPS (Li & Talwalkar, 2019) | | 0.32 | 3360 | 420× | 1.36 | $22.91_{\pm1.65}$ |
| | SETN (Dong & Yang, 2019a) | | 0.32 | 8625 | 1078× | 3.49 | $25.17_{\pm1.68}$ |
| | GDAS (Dong & Yang, 2019b) | | 0.83 | 6965 | 870× | 2.82 | $24.02_{\pm2.75}$ |
| | PC-DARTS (Xu et al., 2020) | | 0.44 | 2844 | 355× | 1.15 | $25.31_{\pm1.38}$ |
| | DrNAS (Chen et al., 2021) | | 0.44 | 6019 | 752× | 2.44 | $26.73_{\pm2.61}$ |
| | **MetaD2A (Ours)** | ✓ | 0.83 | **8** | 1× | **0.003** | $\mathbf{41.50_{\pm4.39}}$ |

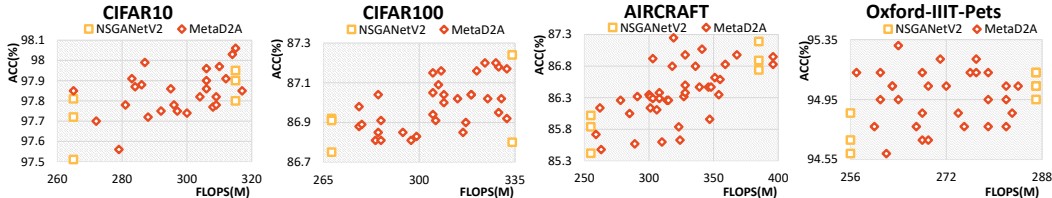

Figure 3: **Performance on Unseen Datasets (Meta-Test)** We show accuracy over flop of both MetaD2A and a transferable NAS referred as to NSGANetV2 (Lu et al., 2020) after meta-training MetaD2A on source database consisting of subsets of ImageNet-1K and architectures in MobileNetV3 search space. Note that each plot point is searched within 125 GPU seconds by MetaD2A.

### 4.2.2 RESULTS ON UNSEEN DATASETS

We search and evaluate the architecture multiple times with both NSGANetV2 and ours on four unseen datasets such as CIFAR-10, CIFAR-100, Aircraft, and Oxford-IIIT Pets with different random seeds. Search times of MetaD2A for CIFAR-10, CIFAR-100, Aircraft, and Oxford-IIIT Pets are within 57, 195, 77, and 170 GPU seconds on average with a single Nvidia RTX 2080ti GPU respectively, while NSGANetV2 needs 1 GPU day with 8 1080ti GPUs on each dataset, which is 5,523 times slower than MetaD2A. Besides the huge speed up, Figure 3 shows that our model can search for a comparable architecture to the NSGANetV2 over flops without a performance drop. Interestingly, even we use naive flop filtering and NSGANetV2 uses an objective function for flop constraints, MetaD2A performs consistently comparably to NSGANetV2 over the different flops. Overall, the results demonstrate that our model also can generalize to unseen datasets not only under the NAS-Bench-201 space but also under a larger MobileNetV3 space with its meta-knowledge.

### 4.3 COMPARISON WITH META-NAS APPROACHES

We further compare our method against Meta-NAS methods (Kim et al., 2018; Elsken et al., 2020; Lian et al., 2019; Shaw et al., 2019) on few-shot classification tasks, which are the main setting existing Meta-NAS methods have been consider. Following (Elsken et al., 2020; Lian et al., 2019), we adopt bi-level optimization (e.g., MAML framework) to meta-learn initial weights of

| Method | NAS | Params (K) | MiniImageNet 5way 1shot | 5way 5shot |
|---|---|---|---|---|
| MAML (Finn et al., 2017) | | 32.9 | 48.70 | 63.11 |
| MAML++ (Antoniou et al., 2018) | | 32.9 | 52.15 | 68.32 |
| AutoMeta (Kim et al., 2018) | ✓ | 28 | 49.58 | 65.09 |
| BASE (Shaw et al., 2019) | ✓ | 1200 | - | 66.20 |
| T-NAS++ (Lian et al., 2019) | ✓ | 26.5 | 54.11 | 69.59 |
| MetaNAS (Elsken et al., 2020) | ✓ | 30 | 49.7 | 62.1 |
| **MetaD2A (Ours)** | ✓ | 28.9 | **54.71** | **70.59** |

Table 2: **Performance on Few-shot Classification Task**

neural architectures searched by our model on a meta-training set of mini-imagenet. As shown in the Table 2, the few-shot classification results on MiniImageNet further clearly show the MetaD2A's effectiveness over existing Meta-NAS methods, as well as the conventional meta-learning methods without NAS (Finn et al., 2017; Antoniou et al., 2018).

### 4.4 EFFECTIVENESS OF METAD2A

Now, we verify the efficacy of each component of MetaD2A with further analysis.

**Ablation Study on MetaD2A** We train different variations of our model on the subsets of ImageNet-1K, and test on CI-FAR10, CIFAR100, and Aircraft in Table 3 with the same experimental setup as the main experiments in Table 1. The MetaD2A generator without the performance predictor (Generator only) outper-

| Model | G | P | Target Dataset CIFAR10 | CIFAR100 | Aircraft |
|---|---|---|---|---|---|
| Random Sampling | | | $93.06_{\pm 0.55}$ | $69.94_{\pm 1.21}$ | $38.15_{\pm 0.99}$ |
| Generator only | ✓ | | $93.96_{\pm 0.22}$ | $71.54_{\pm 0.63}$ | $53.45_{\pm 3.27}$ |
| Predictor only | | ✓ | $93.70_{\pm 0.32}$ | $72.33_{\pm 0.88}$ | $53.39_{\pm 3.13}$ |
| **MetaD2A** | ✓ | ✓ | $\mathbf{94.37_{\pm 0.03}}$ | $\mathbf{73.51_{\pm 0.00}}$ | $\mathbf{58.43_{\pm 1.18}}$ |

Table 3: **Ablation Study of MetaD2A on Unseen Datasets**

forms the simple random architecture sampler (Random Sampling), especially by 15.3% on Aircraft, which demonstrates the effectiveness of MetaD2A over the random sampler. Also, we observe that combining the meta-performance predictor to the random architecture sampler (Predictor only) enhances the accuracy of the final architecture on all datasets. Finally, MetaD2A combined with the performance predictor (MetaD2A) outperforms all baselines, especially by 20.28% on Aircraft, suggesting that our MetaD2A can output architectures that are more relevant to the given task.

**Effectiveness of Set-to-Architecture Generator**

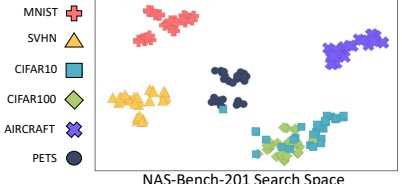

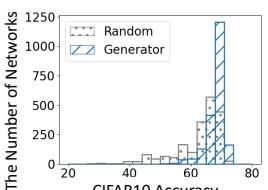

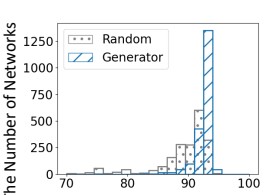

Figure 4: **T-SNE vis. of Latent Space**       Figure 5: **The Quality of Generated Architectures**

We first visualize cross-modal latent embeddings $\{z\}$ of unseen datasets encoded by the meta-

learned generator with T-SNE in Figure 4. Each marker indicates $\{\mathbf{z}\}$ of the sampled subsets of each dataset with different seeds. We observe that the generator classifies well embeddings $\{\mathbf{z}\}$ by datasets in the latent space while clusters $\mathbf{z}$ of the subset of the same dataset. Furthermore, we investigate the quality of generated architectures from those embeddings $\{\mathbf{z}\}$. In the Figure 5, the generator sample 2000 architecture candidates from the embeddings encoded each target dataset and computes the validate accuracy of those architectures. The proposed generator generates more high-performing architectures than the simple random architecture sampler for each target dataset. These results are consistent with Table 3, where the generator (Generator only) enhances the performance compared with the simple random architecture sampler (Random Sampling) consistently on CIFAR10 and CIFAR100. The meta-learned generator allows us to effective and efficient search by excluding the poor-performing architectures of broad search space. We believe the generator replaces the random sampling stage of other NAS methods. We leave to valid it as the future work.

**Could the Generator Create Novel Architectures?**
Since the generator maps set-architecture pairs in the **continuous** latent space, it can generate novel architectures in the meta-test, which are not contained in the source database. To validate it, we evaluate generated 10,000 neural architecture samples of both

| Search Space | Validity | Uniqueness | Novelty |
|---|---|---|---|
| NAS-Bench-201 | 1.0000 | 0.3519 | 0.6731 |
| MobileNetV3 | 0.9831 | 1.0000 | 1.0000 |

Table 4: **Analysis of Generated Architectures**

search space with the measures *Validity*, *Uniqueness*, and *Novelty* following (Zhang et al., 2019) in Table 4. Each is defined as how often the model can generate valid neural architectures from the prior distribution, the proportion of unique graphs out of the valid generations, and the proportion of valid generations that are not included in the training set, respectively. For NAS-Bench-201 search space and MobileNetV3 search space, respectively, the results show the meta-learned generator can generate 67.31%/100% new graphs that do not belong to the training set and can generate 35.19%/100% various graphs, not picking always-the-same architecture seen of the source database.

**Effectiveness of Meta-Predictor** We first demonstrate the necessity of set encoding to handle multiple datasets with a single predictor. In Table 5, we meta-train all models on the source database of NAS-Bench-201 search space and measure Pearson correlation coefficient on the validation tasks (400 unseen tasks) of the source database. Pearson correlation co-

| Predictor | Input Type | | Pearson |
|---|---|---|---|
| Model | Data | Graph | Corr. Coeff. |
| Graph Encoder (GE) Only | | ✓ | 0.6439 |
| DeepSet (Zaheer et al., 2017) + GE | ✓ | ✓ | 0.7286 |
| SetTransformer (Lee et al., 2019a) + GE | ✓ | ✓ | 0.7744 |
| Statistical Pooling (Lee et al., 2020) + GE | ✓ | ✓ | 0.7796 |
| **The Proposed Set Encoder + GE (Ours)** | ✓ | ✓ | **0.8085** |

Table 5: **Effectiveness of Set Encoding to Accurately Predict the Accuracy of Multiple Datasets**

efficient is the linear correlation between the actual performance and the predicted performance (higher the better). Using both the dataset and the computational graph of the target architecture as inputs, instead of using graphs only (Graph Encoder Only), clearly leads to better performance to support multiple datasets. Moreover, the predictor with the proposed set encoder clearly shows a higher correlation than other set encoders (DeepSet (Zaheer et al., 2017), SetTransformer (Lee et al., 2019a), and Statistical Pooling (Lee et al., 2020)).

## 5 CONCLUSION

We proposed a novel NAS framework, MetaD2A (Meta Dataset-to-Architecture), that can output a neural architecture for an unseen dataset. The MetaD2A generator learns a dataset-to-architecture transformation over a database of datasets and neural architectures by encoding each dataset using a set encoder and generating each neural architecture with a graph decoder. While the model can generate a novel architecture given a new dataset in an amortized inference, we further learn a meta-performance predictor to select the best architecture for the dataset among multiple sampled architectures. The experimental results show that our method shows competitive performance with conventional NAS methods on various datasets with very small search time as it generalizes well across datasets. We believe that our work is a meaningful step for building a practical NAS system for real-world scenarios, where we need to handle diverse datasets while minimizing the search cost.

**Acknowledgements** This work was conducted by Center for Applied Research in Artificial Intelligence (CARAI) grant funded by DAPA and ADD (UD190031RD).

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

# A  DETAILS OF THE GENERATOR

## A.1  GRAPH DECODING

To generate the $i^{th}$ node $v_i$, we compute the operation type $\mathrm{o}_{v_i} \in \mathbb{R}^{1 \times n_o}$ over $n_{\mathrm{o}}$ operations based on the current graph state $\mathbf{h}_{\mathcal{G}} := \mathbf{h}_{v_{i-1}}$ and then predict whether the edge exists between the node $v_i$ and other existing nodes. Following (Zhang et al., 2019), when we compute the edge probability $e_{\{v_j,v_i\}}$, we consider nodes $\{v_j | j = i-1, ..., 1\}$ in the reverse order to reflect information from nodes close to $v_i$ to the root node when deciding whether edge connection. Note that the proposed process guarantees the generation of directed acyclic graph since directed edge is always created from existing nodes to a new node.

The graph decoder starts from an initial hidden state $\mathbf{h}_{v_0} = \mathrm{NN}_{\mathrm{init}}(\mathbf{z})$, where $\mathrm{NN}_{\mathrm{init}}$ is an MLP followed by $tanh$. For $i^{th}$ node $v_i$ according to *topological order*, we compute the probability of each operation type $\mathrm{o}_{v_i} \in \mathbb{R}^{1 \times n_o}$ over $n_{\mathrm{o}}$ operations, given the current graph state as the last hidden node $\mathbf{h}_{\mathcal{G}} := \mathbf{h}_{v_i}$. That is, $\mathrm{o}_{v_i} = \mathrm{NN}_{\mathrm{node}}(\mathbf{h}_{\mathcal{G}})$, where $\mathrm{NN}_{\mathrm{node}}$ is an MLP followed by *softmax*. When the predicted $v_i$ type is the end-of-graph, we stop the decoding process and connect all leaf nodes to $v_i$. Otherwise we update hidden state $\mathbf{h}_{v_i}^{(t)}$ at time step $t$ as follows:

$$\mathbf{h}_{v_i}^{(t+1)} = \mathrm{UPDATE}(i, \boldsymbol{m}_{v_i}^{(t)})$$
$$\text{where } \boldsymbol{m}_{v_i}^{(t)} = \sum_{u \in \mathcal{V}_{v_i}^{in}} \mathrm{AGGREGATE}(\mathbf{h}_u^{(t)}) \tag{6}$$

The function UPDATE is a gated recurrent unit (GRU) (Cho et al., 2014), $i$ is the order of $v_i$, and $\boldsymbol{m}_{v_i}^{(t)}$ is the incoming message to $v_i$. The function AGGREGATE consists of mapping and gating functions with MLPs, where $\mathcal{V}_{v_i}^{in}$ is a set of predecessors with incoming edges to $v_i$. For all previously processed nodes $\{v_j | j = i-1, ..., 1\}$, we decide whether to link an edge from $v_j$ to $v_i$ by sampling the edge based on edge connection probability $e_{\{v_j,v_i\}} = \mathrm{NN}_{\mathrm{edge}}(\mathbf{h}_j, \mathbf{h}_i)$, where $\mathrm{NN}_{\mathrm{edge}}$ is a MLP followed by $sigmoid$. We update $\mathbf{h}_{v_i}$ by Eq. (6) whenever a new edge is connected to $v_i$. For meta-test, we select the operation with the max probability for each node and edges with $e_{\{v_j,v_i\}} > 0.5$.

## A.2  META-TRAINING OBJECTIVE

We meta-learn the model using Eq. (1). The expectation of the log-likelihood $\mathbb{E}_{\mathbf{z} \sim q_\phi(\mathbf{z}|\mathcal{D})}\big[\log p_{\boldsymbol{\theta}}\big(\mathcal{G}|\mathbf{z}\big)\big]$ of (2) can be rewritten with negative cross-entropy loss $-L_{CE}^\tau$ for nodes and binary cross-entropy loss $-L_{BCE}^\tau$ for edges, and we slightly modify it using the generated set-dependent graph $\tilde{\mathcal{G}}$ and the ground truth graph $\mathcal{G}$ as the input as follows:

$$-\sum_{i \in \mathcal{V}} \left\{ L_{CE}^\tau\big(\tilde{o}_i, o_i\big) + \sum_{j \in \mathcal{V}_i} L_{BCE}^\tau\big(\tilde{e}_{\{j,i\}}, e_{\{j,i\}}\big) \right\} \tag{7}$$

We substitute the log-likelihood term of Eq. (2) such as Eq. (7) and learn the proposed generator by maximizing the objective (1) to learn $\phi, \boldsymbol{\theta}$, which are shared across all tasks.

# B  GRAPH ENCODING OF THE SET-DEPENDENT PREDICTOR

For a given graph candidate $\mathcal{G}$, we sequentially perform message passing for nodes from the predecessors following the *topological order* of the DAG $\mathcal{G}$. We iteratively update hidden states $\mathbf{h}_{v_i}^{(t)}$ using the Eq. (8) by feeding in its predecessors' hidden states $\{u \in \mathcal{V}_{v_i}^{in}\}$.

$$\mathbf{h}_{v_i}^{(t+1)} = \mathrm{UPDATE}(\boldsymbol{y}_{v_i}, \boldsymbol{m}_{v_i}^{(t)})$$
$$\text{where } \boldsymbol{m}_{v_i}^{(t)} = \sum_{u \in \mathcal{V}_{v_i}^{in}} \mathrm{AGGREGATE}(\mathbf{h}_u^{(t)}) \tag{8}$$

For starting node $v_0$ which the set of predecessors is the empty, we output the zero vector as the hidden state of $v_0$. We use the last hidden states of the ending node as the output of the graph

encoder $\mathbf{h}_f$. Additionally, we exploit Bi-directional encoding (Zhang et al., 2019) which reverses the node orders to perform the encoding process. In this case, the final node becomes the starting point. Thus, the backward graph encoder outputs $\mathbf{h}_b$, which is the last hidden states of the starting node. We concatenate the outputs $\mathbf{h}_f$ of the forward graph encoder and $\mathbf{h}_b$ of the backward graph encoder as the final output of the Bi-directional graph encoding.

## C  BUILDING BLOCKS OF THE SET ENCODER

We use Set Attention Block (SAB) and Pooling by Multi-head Attention (PMA) (Lee et al., 2019a), where the former learns the features for each element in the set using self-attention while the latter pools the input features into $k$ representative vectors. Set Attention Block (SAB) is an attention-based block, which makes the features of all of the instances in the set reflect the relations between itself and others such as:

$$\text{SAB}(\boldsymbol{X}) = \text{LN}(\boldsymbol{H} + \text{MLP}(\boldsymbol{H}))$$
$$\text{where} \quad \boldsymbol{H} = \text{LN}(\boldsymbol{X} + \text{MH}(\boldsymbol{X}, \boldsymbol{X}, \boldsymbol{X})) \tag{9}$$

where LN and MLP is the layer normalization (Ba et al., 2016) and the multilayer perceptron respectively, and $\boldsymbol{H} \in \mathbb{R}^{n_{B_c} \times \mathrm{d}_H}$ is computed with multi-head attention $\text{MH}(\boldsymbol{Q}, \boldsymbol{K}, \boldsymbol{V})$ (Vaswani et al., 2017) which queries, keys, and values are elements of input set $\boldsymbol{X}$.

Features encoded from the SAB layers can be pooled by PMA on learnable seed vectors $\boldsymbol{S} \in \mathbb{R}^{k \times \mathrm{d}_{\boldsymbol{S}}}$ to produce $k$ vectors by slightly modifying $\boldsymbol{H}$ calculation of Eq. (9):

$$\text{PMA}(\boldsymbol{X}) = \text{LN}(\boldsymbol{H} + \text{MLP}(\boldsymbol{H}))$$
$$\text{where} \quad \boldsymbol{H} = \text{LN}(\boldsymbol{X} + \text{MH}(\boldsymbol{S}, \text{MLP}(\boldsymbol{X}), \text{MLP}(\boldsymbol{X}))) \tag{10}$$

While $k$ can be any size (i.e. $k$=1,2,10,16), we set $k = 1$ for generating the single latent vector. For extracting consistent information not depending the order and the size of input elements, encoding functions should be constructed by stacking *permutation-equivariant* layers E, which satisfies below condition for any permutation $\pi$ on a set $\boldsymbol{X}$ (Zaheer et al., 2017):

$$\text{E}(\{\mathrm{x}|\mathrm{x} \in \pi\boldsymbol{X}\}) = \pi\text{E}(\{\mathrm{x}|\mathrm{x} \in \boldsymbol{X}\}) \tag{11}$$

Since all of the components in SAB and PMA are row-wise computation functions, SAB and PMA is permutation equivarint by definition Eq. (11).

## D  SEARCH SPACE

Following the NAS-Bench-201 (Dong & Yang, 2020), We explore the search space consisting of 15,625 possible cell-based neural architectures for all experiments. Macro skeleton is stacked with one stem cell, three stages consisting of 5 cells for each, and a residual block (He et al., 2016) between stages. The stem cell consists of 3-by-3 convolution with 16 channels and cells of the first, second and third stages have 16, 32 and 64, respectively. Residual blocks have convolution layer with the stride 2 for down-sampling. A fully connected layer is attached to the macro skeleton for classification. Each cell is DAG which consists of the fixed 4 nodes and the fixed 6 edge connections. For each edge connection, NAS models select one of 5 operation candidates such as zerorize, skip connection, 1-by-1 convolution, 3-by-3 convolution, and 3-by-3 average pooling. To effectively encode the operation information as the node features, we represent edges of graphs in NAS-Bench-201 as nodes, and nodes of them as edges. Additionally, we add a starting node and an ending node to the cell during training. All nodes which have no predecessors (suceessors) are connected to the starting (ending) node, which we delete after generating the full neural architectures.

## E  EXPERIMENTAL SETUP

### E.1  DATASET

**1) CIFAR-10** (Krizhevsky et al., 2009): This dataset is a popular benchmark dataset for NAS, which consists of $32 \times 32$ colour images from 10 general object classes. The training set consists of 50K images, 5K for each class, and the test set consists of 10K images, 1K for each class. **2) CIFAR-100** (Krizhevsky et al., 2009): This dataset consists of colored images from 100 fine-grained general

object classes. Each class has 500/100 images for training and test, respectively. **3) MNIST** (LeCun & Cortes, 2010): This is a standard image classification dataset which contains 70K 28×28 grey colored images that describe 10 digits. We upsample the images to 32×32 pixels to satisfy the minimum required pixel size of the NAS-Bench 201 due to the residual blocks in the macro skeleton. We use the training/test split from the original dataset, where 60K images are used for training and 10K are used for test. **4) SVHN** (Netzer et al., 2011): This dataset consists of 32×32 color images where each has a digit with a natural scene background. The number of classes is 10 denoting from digit 1 to 10 and the number of training/test images is 73257/26032, respectively. **5) Aircraft** (Maji et al., 2013) This is fine-grained classification benchmark dataset containing 10K images from 30 different aircraft classes. We resize all images into 32×32. **6) Oxford-IIIT Pets** (Parkhi et al., 2012) This dataset is for fine-grained classification which has 37 breeds of pets with roughly 200 instances for each class. There is no split file provided, so we use the 85% of the dataset for training and the other 15% are as a test set. We also resize all images into 32×32. For **CIFAR10** and **CIFAR100**, we used the training, validation, and test splits from the NAS-Bench-201, and use random validation/test splits for **MNIST**, **SVHN**, **Aircraft**, and **Oxford-IIIT Pets** by splitting the test set into two subsets of the same size. The validation set is used to update the searching algorithms as a supervision signal and the test set is used to evaluate the performance of the searched architectures.

### E.2 BASELINES

We now briefly describe the baseline models and our MetaD2A model. **1) ResNet** (He et al., 2016) This is a convolutional network which connects the output of previous layer as input to the current layer. It has achieved impressive performance on many challenging image tasks. We use ResNet56 in all experiments. **2) REA** (Real et al., 2019) This is an evolutional-based search method by using aging based tournament selection, showing evolution can work in NAS. **3) RS** (Bergstra & Bengio, 2012) This is based on random search and we randomly samples architectures until the total time of training and evaluation reaches the budget. **4) REINFORCE** (Williams, 1992) This is a RL-based NAS. We reward the model with the validation accuracy after 12 epochs of training. **5) BOHB** (Falkner et al., 2018) This combines the strengths of tree-structured parzen estimator based baysian optimization and hyperband, performing better than standard baysian optimization methods. **5) RSPS** (Li & Talwalkar, 2019) This method is a combination of random search and weight sharing, which trains randomly sampled sub-graphs from weight shared DAG of the search space. The method then selects the best performing sub-graph among the sampled ones as the final neural architecture. **6) SETN** (Dong & Yang, 2019a) SETN is an one-shot NAS method, which selectively samples competitive child candidates by learning to evaluate the quality of the candidates based on the validation loss. **7) GDAS** (Dong & Yang, 2019b) This is a Gumbel-Softmax based differentiable neural architecture sampler, which is trained to minimize the validation loss with the architecture sampled from DAGs. **8) PC-DARTS** (Xu et al., 2020) This is a gradient-based NAS which partially samples channels to apply operations, to improve the efficiency of NAS in terms of memory usage and search time compared to DARTS. We exploit the code at `https://github.com/yuhuixu1993/PC-DARTS`. **9) DrNAS** (Chen et al., 2021) This is a NAS approach that introduces Dirichlet distribution to approximate the architecture distribution, to enhance the generalization performance of differentiable architecture search. We use the code at `https://github.com/xiangning-chen/DrNAS`. We report the results on **CIFAR10** and **CIFAR100** in this paper using the provided code from the authors on the split set of NAS-Bench 201 while their reported results in the paper of the authors are 94.37 and 73.51, respectively on random training/test splits on **CIFAR10** and **CIFAR100**. **10) MetaD2A (Ours)** This is our meta-NAS framework described in section 3, which can stochastically generate task-dependent computational graphs from a given dataset, and use the performance predictor to select the best performing candidates. We follow the same settings of NAS-Bench-201 (Dong & Yang, 2020) for all baselines and use the code at `https://github.com/D-X-Y/AutoDL-Projects` except for 8), 9) and 10).

## E.3 IMPLEMENTATION DETAILS

| Hyperparameter | Value |
|---|---|
| The number of inputs of class b | 20 |
| Dimension of $\mathbf{v}_c$ $d_{\mathbf{v}_c}$ | 56 |
| Dimension of $\mathbf{h}_e$ $d_{\mathbf{h}_e}$ | 56 |
| Dimension of $\mathbf{S}$ $d_{\mathbf{S}}$ | 56 |
| Dimension of $\mathbf{z}$ $d_{\mathbf{z}}$ | 56 |
| Dimension of $\mathbf{h}_{v_i}$ for generator | 56 |
| Dimension of $\mathbf{h}_{v_i}$ for predictor | 512 |
| The number of operators $n_o$ | 5 |
| Learning rate | 1e-4 |
| Batch size | 32 |
| KL-divergence weighting value $\lambda$ | 5e-3 |
| Training epoch | 400 |

Table 6: Hyperparameter setting of MetaD2A on NAS-Bench-201 Search Space

We use embedding features as inputs of the proposed set encoder instead of raw images, where the embedding features are generated by ResNet18 (He et al., 2016) pretrained with ImageNet-1K (Deng et al., 2009). We adopt the teacher forcing training strategy (Jin et al., 2018), which performs the current decoding process after correcting the decoded graph as the true graph until the previous step. This strategy is only used during meta-training and we progress subsequent generation based on the currently decoded graph part without the true graph information in the meta-test. We use mini-batch gradient descent to train the model with Eq. (1). The values of hyperparameters which we used for both MetaD2A generator and predictor in this paper are described in Table 6. To train searched neural architectures for all datasets, we follow the hyperparameter setting of NAS-Bench-201 (Dong & Yang, 2020), which is used for training searched neural architectures on CIFAR10 and CIFAR100. While we report accuracy after training 50 epoch for MNIST, the accuracy of 200 epoch are reported for all datasets except MNIST.

