# OpenReview forum: "Rapid Neural Architecture Search by Learning to Generate Graphs from Datasets"
_ICLR.cc/2021/Conference — ICLR 2021 Poster_

### Official Review · AnonReviewer1 · 2020-10-28
**The authors present a novel framework to learn architecture efficiently from datasets. Even though some components have been proposed before, the way of combination is interesting to me. However, their experiment results are not convincing.**

**Rating:** 5
**Confidence:** 4

**Review:**

The authors propose a novel framework named MetaD2A. Their motivation is interesting to me and they clarify their difference compared with traditional metaNAS in Figure 1. There are mainly three components in MetaD2A: a set encoder, a graph decoder and a meta-performance predictor. I think the main contribution of this paper is their intuition that performing neural architecture search rapidly from the datasets, while the components are all proposed before in different NAS scenarios. In summary, the paper has following drawbacks that need to be further concerned:

1. Since the predictor consists of two linear layers, we cannot take architecture with different nodes as input. Thus, it will limit the generalization ability of the whole algorithm. Moreover, there are also different kinds of performance predictors in NAS field like LSTM and GCN predictor [1]. And I prefer to see the effect of predictor part with different predictors.

2. In Table 1, MetaD2A is pretrained on Meta-ImageNet while other baselines are trained from the scratch. I think this is unfair to compare them. It's better to compare MetaD2A with some MetaNAS methods listed in related work part such as [2]. In my view, I think MetaD2A pays more emphasis on meta-learning in NAS field.

2. In ablation study section, they only compare MetaD2A with random, which is a somewhat weak baseline. However, the improvement of their method is not significant. I think there are some other perspectives to perform ablation study. For instance, they can replace hierarchical set pooling with flatten set pooling and then see the importance of different component.

[1] Shi, Han, et al. "Bridging the Gap between Sample-based and One-shot Neural Architecture Search with BONAS." Neural Information Processing Systems. 2020.

[2] Lian, Dongze, et al. "Towards fast adaptation of neural architectures with meta learning." International Conference on Learning Representations. 2019.

---

> ### Author Response · Authors · 2020-11-18
> **Responses to the comment 3 (ablation study)**
>
> **(3) In ablation study section, they only compare MetaD2A with random, which is a somewhat weak baseline. However, the improvement of their method is not significant. I think there are some other perspectives to perform ablation study. For instance, they can replace hierarchical set pooling with flatten set pooling and then see the importance of different component.**
>
> - We **do provide the ablation study in Table 5** for a total of 4 modules including random (random, random with predictor, generator without predictor, generator with predictor). In addition, we perform the ablation study to analyze components of the input type, the encoder, and the decoder in Table 6.
>
> - Moreover, in ablation study of Table 5, our method achieves **22.16%** higher performance over random on aircraft, which we believe is a more than significant improvement.
>
> - We **do provide the ablation study on the flat set encoding in Table 6** (Please see the fourth and the fifth row of Table 6). We can see that the hierarchical set encoding and bi-directional graph encoding both improves the performance of the predictor.
>
> ---
>
> We hope that the above responses clear up your confusion. Please let us know if there is anything else we should provide.

---

> ### Author Response · Authors · 2020-11-18
> **Responses to the comment 2 on the comparison against existing works**
>
> **(2-1) In Table 1, MetaD2A is pretrained on Meta-ImageNet while other baselines are trained from the scratch. I think this is unfair to compare them.**
>
> - Please note that MetaD2A **does not** train on the target datasets at all after it is trained on Meta-ImageNet, while all existing approaches require training on the target datasets. MetaD2A only uses 20 samples per class from each target dataset for inference, **without any training regardless of the number of datasets**we target.
>
> - Thus, when considering the total number of examples to train on, when generating architectures for multiple datasets, if they, in sum, have a larger number of samples than MetaImageNet, the number of samples used by MetaD2A will be less. Moreover, as the number of datasets we consider becomes much larger, our method becomes even more sample-efficient since it only has to be trained once on the source dataset.
>
> - Please note that such a strategy to "train only once and adapt to new datasets at no training cost", is the main reason why MetaD2A can rapidly generate adaptive neural architecture for a given dataset, which is made possible by our **amortized meta-learning framework**.
>
> **(2-2) It's better to compare MetaD2A with some MetaNAS methods listed in related work part such as [2]. In my view, I think MetaD2A pays more emphasis on meta-learning in NAS field.**
>
> - MetaNAS methods listed in the related work target for few-shot classification, and cannot scale to large-scale learning with a large number of dataset-architecture pairs (1,679 and 3,752 tasks for training the generator and the performance predictor), each of which contains $260,000$ examples). One of our main contributions is in the proposal of amortized meta-learning of the cross-modal space of dataset and neural architectures (Page 2), which enables large-scale meta-learning of a NAS framework.
>
> - We further compared our method against Meta-NAS methods [3,4,5] on few-shot classification tasks (5 way 5 shot and 5 way 1 shot classifications on mini-imagenet), which is the main setting existing Meta-NAS methods consider. Note that we can generate an architecture for each task almost instantly with MetaD2A for the few-shot task. Following [4, 5], we adopt bi-level optimization (e.g., MAML framework) to learn initial weight of the searched architectures on a meta-training set of mini-imagenet. The results are as follow:
>
> |       Method       	| Params (K) 	| 5way 1shot 	| 5way 5shot 	|
> |:------------------:	|:----------:	|:----------:	|:----------:	|
> |        MAML [1]       	|    32.9    	|    48.70   	|    63.11   	|
> |       MAML++ [2]     	|    32.9    	|    52.15   	|    68.32   	|
> |        BASE [3]       	|    1200    	|      -     	|    66.20   	|
> |       T-NAS++ [4]     	|    26.5    	|    54.11   	|    69.59   	|
> |       MetaNAS [5]     	|     30     	|    49.7    	|    62.1    	|
> | **MetaD2A (ours)** 	|    28.9    	|  **54.71** 	|  **70.59** 	|
>
>
> The few-shot classification results on MiniImageNet further clearly show the MetaD2A's effectiveness over existing Meta-NAS methods, as well as conventional meta-learning without NAS.
>
> [1] Finn, C, et al. Model-agnostic meta-learning for fast adaptation of deep networks, ICML17
>
> [2] Antoniou, A et al., How to train your MAML, ICLR19
>
> [3] Shaw, A. et al, Meta Architecture Search, NeruIPS 19
>
> [4] Lian, D. et al., Towards Fast Adaptation of Neural Architectures with Meta Learning, ICLR19
>
> [5] Elsken, T et al., Meta-Learning of Neural Architectures for Few-Shot Learning, CVPR20
>
> ---
>
> We add the discussions for scalability of the proposed method compared with conventional Meta-NAS approaches in Method Section 3.3 and include the above experimental results to Section D.2 of Appendix in the revision. We appreciate R1's helpful comments.

---

> ### Author Response · Authors · 2020-11-18
> **Responses to the comments on the Novelty and the Performance Predictor.**
>
> Thank you for your reviewing efforts and constructive comments. We respond to your comments below:
>
> **I think the main contribution of this paper is their intuition that performing neural architecture search rapidly from the datasets, while the components are all proposed before in different NAS scenarios.**
>
> - Our main contribution, which largely differentiates it from the existing works in NAS, is the proposal of an **amortized meta-learning framework for learning a cross-modal latent space**of the datasets and architectures that allows to generate an architecture for a given dataset **without**training on it. Such meta-learning of the cross-modal embedding space has not been tried by any of the existing NAS approaches, due to the limited scalability of conventional meta-learning frameworks (e.g. MAML), and is made possible only by our amortized meta-learning algorithm.
>
> ---
>
> **(1-1) Since the predictor consists of two linear layers, we cannot take architecture with different nodes as input. Thus, it will limit the generalization ability of the whole algorithm.**
>
> - This is a misunderstanding and the performance predictor **does not only consist of two linear layers**. The two linear layers are placed on top of a dataset encoder and a graph encoder, where the latter can encode neural networks with different architectures. Please see Section 3.2 and concept Figure 1 and 2.
>
> **(1-2) There are also different kinds of performance predictors in NAS field like LSTM and GCN predictor [1].**
>
> - We would like to point out our performance predictor is **meta-learned**on multiple tasks from a single dataset, such that it can **generalize**to the performance prediction of a **novel neural architecture on a novel dataset without training**, which is not possible with any of the existing performance predictors.
>
> - To our knowledge, **the performance predictors proposed in the existing NAS works need re-training for the new dataset** since they have **no module to handle multiple datasets** (They are set-specific predictors that have encoders only for graphs.). On the other hand, our proposed meta-performance predictor consists of dataset encoder and graph encoder to address such a problem with one training. Moreover, the proposed meta-performance predictor learns meta-knowledge accumulated from multiple tasks of MetaImageNet. Thus, the predictor can predict performance for the new unseen dataset. As the number of target datasets increases, **the time complexity of other predictors linearly increases while our predictor has a constant time complexity of O(1).** In addition, none of the existing works in both NAS or meta-learning can handle the scale of meta-learning (training over 1,679 datasets, each of which contains 2,000 samples) as our MetaD2A does.
>
> **(1-3) I prefer to see the effect of predictor part with different predictors.**
>
> - Yet, following R1’s suggestion, we compare our meta-predictor against an LSTM predictor and a GCN predictor, as well as a two-layer linear predictor (to show that this is not our model). We train all of them on the multiple tasks from MetaImageNet, and use them to predict the accuracy on 400 unseen MetaImageNet tasks. We report the performance as the Pearson correlation coefficient between the predicted accuracy and the actuary accuracy (higher the better). The results are given in the table below:
>
> #### **Predictor Correlation**
> |                    | Linear Layer |  LSTM  |   GCN  | MetaD2A-G | MetaD2A-DG |
> |:-----------:   |:------------:       |:------:   |:------:   |:-----------:      |:----------:           |
> | Correlation |    0.0420      | 0.0064 | 0.6076 |   0.6426        | **0.7976** |
>
> - The linear or LSTM-based predictors completely fail to generalize to new tasks, which is expected since they are too simple and ignore the change of accuracy of the same architecture on different datasets. GCN-based predictor works reasonably well, obtaining the Pearson correlation score of 0.6076. Yet, it significantly underperforms MetaD2A-G, which predicts the performance only based on the graph of the neural architecture, without consideration of the dataset. Finally, our full model, MetaD2A-DG which predicts the performance based on both **the dataset and the architecture** achieves the highest Pearson correlation score, significantly outperforming all baselines.
>
> ---
>
> We clarify the contribution of our meta-performance predictor in Section 3.2 of Method and include the above experimental results in Section D.1. of Appendix. Thank you for the constructive suggestion.

---

> ### Author Response · Authors · 2020-11-23
> **Responses and the revision uploaded**
>
> Dear reviewer,
>
> Could you check our responses to your comments as well as the revision that reflects them? We have answered all your questions, provided the experimental results you have requested (comparison with existing predictor and comparison with conventional Meta-NAS approaches), and clarified your concern regarding the novelty over conventional NAS, and ablation study.
>
> Thanks, Authors

---

> ### Author Response · Authors · 2020-11-25
> **The interactive discussion phase ending in less than 7 hours.**
>
> Dear reviewer,
>
> The interactive discussion phase will end in less than 7 hours, and we cannot have discussions with you anymore after the deadline. Could you please check if our responses to your comments and the new experimental results you have requested? Please let us know if there is any other things that we need to clarify or provide. We thank you so much for your helpful and insightful suggestion.
>
> Thank you,
> Authors.

---

### Official Review · AnonReviewer3 · 2020-10-28
**This paper proposes a framework to learn meta knowledge which helps to transfer architecture search task among different datasets. This is a new attempt to adopt meta NAS on such a scene. This work encode dataset into embedding space, then sample a vector from the space and decode it into an architecture. A predictor is also used to find the optimal architecture.**

**Rating:** 4
**Confidence:** 4

**Review:**

Pros:
1.	This paper proposes new scene, where a meta model is pre-trained on some datasets, and transfer the learned meta feature onto other datasets to do fast adoption. This scene can be applied in variety of domains.
2.	The experiment shows this method can fast adapt NAS from one image dataset to others and achieve SOTA performance.
3.	The paper is well-organized, the paper structure is clear.

Cons:
1.	The three parts of your model are of little novelty. The dataset encoding part is just borrowed and there is no improvement to adapt NAS tasks. Similar graph decoder is proposed in previous NAS works [1] and performance predictor are proposed more times. It seems that the proposed framework is just putting existing models together.
2.	No comparing to other methods on fast adaptation by NAS such as [2]. Besides, meta-learning methods may also be compared.
3.	According to Figure 8 and Figure 9, it is likely that your graph decoder can only generate one type of edge connections. Your graph decoder may fail. Since your framework needs other methods (GDAS / NAS-Bench-201) to provide training material. These materials are all good architectures. It is possible that your framework just gives architectures the same as those good architectures rather than using meta features. Your experiment should prove that your model can generate variety of architectures.

Overall Review:
This paper proposes a new scene of fast adaption of NAS, which may be a good direction of NAS & meta-learning. The paper proposes a framework to generate good architectures according to the datasets. However, the model may need to improved and more experiment need to be done to solve the problems mentioned above.

[1]Does unsupervised architecture representation learning help neural architecture search? NeurIPS 20’
[2]Fast neural network adaptation via parameter remapping and architecture search ICLR 20’

---

> ### Author Response · Authors · 2020-11-18
> **Response to the Comment 3 (Correctness and diversity of generated architectures)**
>
> **(3-1) According to Figure 8 and Figure 9, it is likely that your graph decoder can only generate one type of edge connections. Your graph decoder may fail.**
>
> - This is a misunderstanding and our graph decoder did not fail. We use **nodes to represent the operation types**, and thus our graph decoder generates the graph with fixed edges with **variable node types** such as the graphs in Figure 8 and Figure 9. Note that this is different from [Liu et al. 19 and Dong & Yang 20] where the operations are represented with edges. Thus our model did generate diverse neural networks with different operations.
>
>
> **(3-2) Since your framework needs other methods (GDAS / NAS-Bench-201) to provide training material. These materials are all good architectures. It is possible that your framework just gives architectures the same as those good architectures rather than using meta features. Your experiment should prove that your model can generate variety of architectures.**
>
> - Note that our method is a probabilistic approach and can generate multiple architectures even for a **single** dataset, rather than simply retrieving a memorized architecture from the training set. We further performed an additional experiment to show that MetaD2A can generate diverse architectures, instead of retrieving a memorized architecture from the training set. To this end, we generated 10000 neural architecture samples with our MetaD2A on the meta-training set of the MetaImageNet dataset, and measured the quality of the generated architecture by the 1) Validity and 2) Novelty, following [Zhang et al. 19].
>
> - **1) Validity**measures the proportion of the neural architectures that are valid (e.g. graphs with all nodes connected to some other nodes).
> - **2) Novelty**measures the proportion of the valid neural architectures that are never seen in the training set.
>
> - The result in the Table below shows that MetaD2A can generate **diverse and novel**architectures that **never appeared in the training set**.
>
> #### **The Proportion of Novelty Graphs (%)**
> | Samples 	| Validity 	| Novelty 	|
> |:-------:	|:--------:	|:-------:	|
> |  10000  	|   100%   	|  67.31% 	|
>
> [Zhang et al. 19] D-VAE: A Variational autoencoder for directed acyclic graphs. NeurIPS 2019.
>
> ---
> We have included this result in the Experiments Section 4.3 of the revision. We appreciate your constructive comments.

---

> > ### Comment · AnonReviewer3 · 2020-11-24
> > **Response 3**
> >
> > Concern 3: The authors give experiment about the novelty of the generated architectures. But my concern here is the fixed edges of the generated architectures. I think it means the edge decoding part of your model cannot generate different kinds of edge connections, can you show that your model is able to generate different types of edge connection, or you can prove that this type of edge connection is good whatever the node operation is and wherever the model adapts.

---

> > > ### Author Response · Authors · 2020-11-24
> > > **Response to new comment 3 (Diverse edge connection generation)**
> > >
> > > - We appreciate your constructive comments. First, we would like to clarify that the fixed edge is the **property of the NAS-Bench-201 search space** and not the limitation of our method. Since we performs NAS on the NAS-Bench-201 search space, which is the standard benchmark dataset, the generated networks will have fixed edges. We have clarified the description of NAS 201 search space in the **Section B of the Appendix, in the revision**.
> > >
> > > - However, our model **is able to generate graphs with diverse edges**. To address R3's concern, we performed an experiment to show that MetaD2A generates diverse edge connections, and included **the experimental results in D.5 of the Appendix**of the revision. We train MetaD2A in the search space of [Zhang et al. 19], where architectures consist of 6 layers,  which each node represents one of 6 operations (e.g. 3×3 convolutions,  5×5convolutions, 3×3 and 5×5 depthwise-separable convolutions, 3×3 max pooling, and 3×3 average pooling), with diverse edge connections between nodes. In **Figure 10 of Appendix D.5**, we observe that MetaD2A successfully generates architectures with various edge connections including different types of operations of nodes. We hope that this result addresses your concern. We believe that including the results in the paper has further reduced confusion regarding the diversity of the generated graphs.
> > >
> > > [Zhang et al. 19] D-VAE: A Variational autoencoder for directed acyclic graphs. NeurIPS 2019.

---

> > > ### Author Response · Authors · 2020-11-25
> > > **The interactive discussion phase ending in less than 9 hours.**
> > >
> > > Dear reviewer,
> > >
> > > Could you check the results we provide in **Figure 10 of the Appendix D.5**? We have shown that our model can generate graphs with diverse edges from the **DVAE search space**, which shows that the reason we had fixed edges (with only the nodes changing) was because of the**NAS Bench-201 search space**, and is not because of the limitation with our model.
> > >
> > > Could you also please let us know any other things we should clarify or address, since we have only 9 hours before the end of the interactive discussion phase? We thank you for your helpful comments which we believe have largely improved our paper.

---

> ### Author Response · Authors · 2020-11-18
> **Response to the Comment 2 (Missing comparison to [2] and Meta-NAS methods)**
>
> **(2-1) No comparing to other methods on fast adaptation by NAS such as [2].**
>
> - As a conceptual difference, [2] starts from a pretrained seed network and adapts it to a target dataset using a differentiable NAS. Therefore, whenever a new dataset is given, **[2] needs to train on the target dataset** in order to adapt and finetune the architecture, which takes a large amount of training time  (Please see the search time of DARTS and PC-DARTS in Table 1 and 2), while ours can generate the architecture by performing multiple forward steps, that is extremely fast.
>
> - Moreover, as you suggested, we compared our model with [2] on multiple datasets. For FNA, we report the results of the models pre-trained on two datasets, CIFAR-100 (FNA-CIFAR100) and MetaImageNet (FNA-ImageNet) which is also used in MetaD2A. The results are given in the Table below, which shows that MetaD2A **significantly outperforms it** with **orders of magnitude**smaller search time:
>
> #### **Accuracy (%)**
> |    Methods   |   MNIST   |    SVHN   |  CIFAR10  |  CIFAR100 |  AIRCRAFT |    Pets   |
> |:------------:|:---------:|:---------:|:---------:|:---------:|:---------:|:---------:|
> |     DARTS    |   98.82   |   65.71   |   54.30   |   15.61   |   22.50   |   18.14   |
> | FNA-ImageNet |   93.11   |   94.75   |   39.64   |   39.19   |   22.46   |   11.95   |
> | FNA-CIFAR100 |   99.51   |   94.19   |   70.41   |     -     |   26.43   |   13.57   |
> | MetaD2A      | **99.70** | **96.56** | **94.37** | **73.51** | **60.31** | **38.74** |
> #### **Search Time (s)**
> |    Methods   	| MNIST 	|  SVHN 	| CIFAR10 	| CIFAR100 	| AIRCRAFT 	| Pets 	|
> |:------------:	|:-----:	|:-----:	|:-------:	|:--------:	|:--------:	|:----:	|
> |     DARTS    	| 20606 	| 25286 	|   7046  	|   16747  	|   2899   	| 2469 	|
> | FNA-ImageNet 	| 19945 	| 23248 	|   9016  	|   13732  	|   2762   	| 1808 	|
> | FNA-CIFAR100 	| 19945 	| 15169 	|   9462  	|     -    	|   3178   	| 2357 	|
> |    MetaD2A   	|   **32**  	|   **30**  	|    **26**   	|    **68**    	|    **34**   	|  **38**  	|
>
> - While both FNA models achieves better performance over DARTS with reduced search time over it, since FNA **requires to train on each dataset**, it inevitably requires a large amount of computation during the adaptation process. Contrarily, MetaD2A generates a neural architecture for most of each given task with **less than a minute** as it requires only forward passes, **without any training**on the target dataset.
>
> - We include the discussion of [2] in the related work section, and add [2] as a baseline in the main Table 2 and Table 3 of the revision.
>
> ---
>
> **(2-2) Besides, meta-learning methods may also be compared.**
>
> - Conventional Meta-NAS approaches [Lian et al. 19, Shaw et al. 19, Elsken et al. 20] mainly targets few-shot classification, and **cannot meta-learn over a large number of dataset-architecture pairs** since they resort to gradient-based meta-learning, which requires expensive lookahead steps (Please see Figure 1(b)). Overcoming such a limitation of conventional Meta-NAS methods, and meta-learning in general, is one of the main contributions of this work, which we believe is highly novel, in both the perspective of NAS and meta-learning.
>
> - We further compared our method against Meta-NAS methods [3,4,5] on few-shot classification tasks (5 way 5 shot and 5 way 1 shot classifications on mini-imagenet), which is the main setting existing Meta-NAS methods consider. Note that we can generate an architecture for each task almost instantly with MetaD2A for the few-shot task. Following [4, 5], we adopt bi-level optimization (e.g., MAML framework) to learn initial weight of the searched architectures on a meta-training set of mini-imagenet. The results are as follow:
>
> |       Method       	| Params (K) 	| 5way 1shot 	| 5way 5shot 	|
> |:------------------:	|:----------:	|:----------:	|:----------:	|
> |        MAML [1]       	|    32.9    	|    48.70   	|    63.11   	|
> |       MAML++ [2]     	|    32.9    	|    52.15   	|    68.32   	|
> |        BASE [3]       	|    1200    	|      -     	|    66.20   	|
> |       T-NAS++ [4]     	|    26.5    	|    54.11   	|    69.59   	|
> |       MetaNAS [5]     	|     30     	|    49.7    	|    62.1    	|
> | **MetaD2A (ours)** 	|    28.9    	|  **54.71** 	|  **70.59** 	|
>
> The few-shot classification results on MiniImageNet further clearly show the MetaD2A's effectiveness over existing Meta-NAS methods, as well as conventional meta-learning without NAS. We add this result to the Appendix of the revision.
>
> [1] Finn, C, et al. Model-agnostic meta-learning for fast adaptation of deep networks, ICML17
>
> [2] Antoniou, A et al., How to train your MAML, ICLR19
>
> [3] Shaw, A. et al, Meta Architecture Search, NeruIPS 19
>
> [4] Lian, D. et al., Towards Fast Adaptation of Neural Architectures with Meta-Learning, ICLR19
>
> [5] Elsken, T et al., Meta-Learning of Neural Architectures for Few-Shot Learning, CVPR20

---

> > ### Comment · AnonReviewer3 · 2020-11-24
> > **Response 2**
> >
> > Concern 2-1: The authors give experiment results compared with [2], which shows the effectiveness and the efficiency of the proposed model.
> >
> > Concern 2-2: The authors give experiment results in few-shot learning setting and compared with meta learning methods. The results show the proposed model can also be applied in this setting and achieve SOTA performance.

---

> > > ### Author Response · Authors · 2020-11-24
> > > **Response to new comment 2**
> > >
> > > Thank you for your helpful comments. We believe that our paper became much stronger after including the additional experiments you have suggested.

---

> ### Author Response · Authors · 2020-11-18
> **Response to the Comment 1 (Novelty)**
>
> We thank you for your constructive comments. We provide responses to your comments below:
>
> **(1) The three parts of your model are of little novelty. The dataset encoding part is just borrowed and there is no improvement to adapt NAS tasks. Similar graph decoder is proposed in previous NAS works [1] and performance predictor are proposed more times. It seems that the proposed framework is just putting existing models together.**
>
> - Our model consists of a data-to-graph generator and meta-performance predictor (Please refer to concept Figure 2). Both are designed to learn meta-knowledge while to our best knowledge, none of conventional NAS methods design their framework to learn the meta-knowledge using graph encoder/decoder and predictor. Most conventional NAS methods use graph encoder/decoder for graph-to-graph reconstruction, which has a limitation to be re-trained when a target dataset is changed. On the other hand, our data-to-graph generator once learned meta-knowledge with MetaImageNet works for new dataset **without additional training** by taking input dataset. We believe such an **amortized meta-learning framework to learn cross-modal latent space** between datasets and graphs to generate set-suitable architectures is challenging and has novelty.
>
> - To our knowledge, the performance predictors proposed in the existing NAS works need re-training for the new dataset since they have no module to handle multiple datasets (Most of them have encoders only for graphs). On the other hand, our proposed meta-performance predictor consists of dataset encoder and graph encoder to address such a problem with one training. Moreover, the proposed meta-performance predictor learns meta-knowledge accumulated from multiple tasks of MetaImageNet. Thus, the predictor can predict performance for the new unseen dataset. As the number of target datasets increases, the time complexity of other predictors linearly increases while our predictor has a constant time complexity of O(1). In addition, none of the existing works in both NAS or meta-learning can handle the scale of meta-learning (training over **1,679 datasets**, each of which contains **2,000 samples**) as our MetaD2A does.
>
> - Moreover, the dataset encoding part is **new**, although we borrow the components from the Set Transformer [Lee et al. 19] or the hierarchical set encoder from [Lee et al. 20], since the former does not have the notion of hierarchical encoding and the latter resort to simple averaging of the instance-wise representations.
>
> ---
>
> We have also responded to your second and the third question in the responses below, which provide the results of the additional experiments you requested. We hope that the discussions and additional experiments in this response further strengthens our argument and addresses R3's concerns. We include the experiments of (2-1) and (3-2) in the Experiments section and (2-2) in the Appendix of the revision. Please let us know any more things we should provide.

---

> > ### Comment · AnonReviewer3 · 2020-11-24
> > **Response 1**
> >
> > Concern 1: The authors address that although similar components have appeared in previous work, they do not contain meta mechanism can be applied on such a scene where new datasets are not trained. The contribution of this model is adding this mechanism. I think it is of novelty and OK.

---

> > > ### Author Response · Authors · 2020-11-24
> > > **Response to new comment 1**
> > >
> > > Thank you for appreciating the novelty of our work. We believe that the proposed method makes important contributions in both NAS and meta-learning fields.
> > >
> > > - Our model is the first NAS method that can rapidly search for architectures for **novel (unseen) datasets without training on the target dataset** (only with **forward passes**) rapidly by transferring meta-knowledge learned over large number of tasks sampled from the source dataset.
> > >
> > > - Moreover, such meta-learning is possible with our **scalable amortized meta-learning framework** which can meta-learn over **larger number of datasets**with **large number of samples** without any gradient update steps.

---

> ### Author Response · Authors · 2020-11-23
> **Responses and Revision uploaded**
>
> Dear reviewer,
>
> Could you check our responses to your comments as well as the revision that reflects them? We have clarified the novelty, clearly described the correctness of the generated architectures, provided the results on the diversity of the generated architectures you suggested, and included the experiments for comparison against Meta-NAS approaches and FNA. We would like your feedback since we cannot interact with you after this Tuesday, which is the end of the interactive discussion phase.
>
> Thanks, Authors

---

### Official Review · AnonReviewer2 · 2020-10-29
**Solid paper, weak accept**

**Rating:** 6
**Confidence:** 2

**Review:**

The authors address neural architecture search (NAS) scenarios. In particular, a framework, MetaD2A, is proposed, which yields a neural architecture for a new dataset. In a nutshell, the framework learns a "dataset-to-neural-network-architecture" transformation using a database of datasets and architectures. Each dataset is encoded via a "set encode" and the architecutres are obtained via a "graph decoder". The experiments demonstrate the usefullness of the approach and its improvements over conventual NAS approaches. The approach could be described

Positive:
- The results look very solid and indiciate improvements (time/prediction performance) over existing approaches
- The paper is well written and structured
- Additional details (e.g., implementation details) are provided in the appendix

Negative:
- The authors claim that NAS with meta learning has only been done with small datasets in the past. However, the authors do not really use big datasets as well (see Appendix C; the ImageNet subset considered is small as well, if I understand this correctly)

To sum up, I think the work might be a suitable candidate for being accepted at ICLR. I have to admit that this is not precisely an area of my expertise, so I might be missing something.

---

> ### Author Response · Authors · 2020-11-10
> **Initial Response**
>
> Thank you for your constructive comments. We address your comments below:
>
> **Negative: The authors claim that NAS with meta-learning has only been done with small datasets in the past. However, the authors do not really use big datasets as well (see Appendix C; the ImageNet subset considered is small as well, if I understand this correctly)**
>
> - Existing Meta-NAS approaches [Lian et al. 19, Shaw et al. 19, Elsken et al. 20] cannot scale to such a large task with a large number of samples, since they rely on gradient-based meta-learning with rollout gradient steps. For few-shot learning, taking a few gradient steps is more than enough since there are only a few samples from which we can compute the gradients, but for large-scale learning, taking few gradient steps with minibatch sampling will only consider a very small portion of the samples in the task, which will be largely insufficient for the model to converge. Thus the model needs to take a large number of gradient steps with large tasks (with a large number of samples). However, this is extremely costly when we want to train over a large number of tasks (we consider **1,679 tasks**to train the MetaD2A generator, and **3,752 tasks**for the meta-performance predictor), and is almost infeasible when computing second-order derivatives as in MAML.
>
> - On the contrary, MetaD2A can handle the large-way many-shot problem tackled by general classification tasks, since it does not require any rollout gradient steps when inferring the architecture. For example, CIFAR-100 in Table 2 is orders of magnitudes larger than the tasks considered in a few-shot classification (m-way k-shot classification) existing Meta-NAS methods target, where the number of samples is $m\times{k}$. For conventional settings (5-way 5-shot), the number of samples per task is 25. Contrarily, the number of samples of CIFAR-100 used is **100$\times$20=2,000**, which is **$80$ times larger**than the tasks in 5-way 5-shot classification. Further, the number of instances(e.g. 20) can be set to a much larger number.
>
> Please let us know if there is anything else that you want us to clarify or provide.

---

### Author Response · Authors · 2020-11-18
**Summary of the updates in the revision**

We thank the reviewers for constructive comments. We appreciate that the reviewers consider the proposed framework to be new attempt (R1, R3), interesting (R1), good direction of both NAS and meta-learning (R3), and the results to be solid (R2). The reviewers also mention that the paper is well written (R2) and has clear structure (R2, R3). We have updated the paper by making the following changes:

---

- We revised the paper to clarify the technical novelty of meta-predictor and included **the experiments compared with the competitive predictors** in Appendix of the paper. (**R1**)

---

- We clarified the ablation study in Experiment Section 4.3. (**R1**)

---

- We added **the discussions for scalability of the proposed method compared with conventional Meta-NAS approaches** in Method Section 3.3. (**R2** and **R1**)

---

- We included **the experiments to show that our model can generate novel architectures** in Experiment Section 4.3. (**R3**)

---

- We included **the experiments for comparing MetaD2A with FNA [1]** which studies fast adaptation by NAS in Table 2 and Table 3. (**R3**)

---

- We added the discussions on FNA[1] in the Related Work Section. (**R3**)

---

- We performed **the experiments to compare our model and Meta-NAS methods** and included the results in Appendix of the paper. (**R1** and **R3**)

---

[1] Fast neural network adaptation via parameter remapping and architecture search. ICLR 20

---

We strongly believe that the problem we tackle (rapid architecture search without training for new task) and the idea (amortized meta-learning framework to learn cross-modal latent space) we propose to tackle the problem are both highly novel, and provide important contributions to the research in both meta-learning and NAS. We also believe that the new experimental results of the revision effectively address the concerns of the reviewers on comparison with Meta-NAS approaches, and diversity of generated architectures.

---

### Author Response · Authors · 2020-11-23
**The end of the discussion phase approaching**

Dear Reviewers,

Could you please go over our responses and the revision since we can have interactions with you only by this Tuesday (24th)? We have responded to your comments and faithfully reflected them in the revision, and provided additional experimental results that you have requested. We sincerely thank you for your time and efforts in reviewing our paper, and your insightful and constructive comments.

Thanks, Authors

---

### Decision · Program_Chairs · 2021-01-07
**Final Decision**

**Decision:**

Accept (Poster)

**Comment:**

The authors proposed a meta learning framework for NAS, namely MetaD2A (Meta Dataset-to-Architecture), that can stochastically generate graphs (architectures) from a given set (dataset) via a dataset-architecture latent space learned with amortized meta-learning. Each dataset is encoded via a set encoder and the architecutres are obtained via a graph decoder. MetaD2A is trained once on a database consisting of datasets and pretrained networks and can rapidly search a neural architecture for a novel dataset. While the set encoder and graph decoder for NAS have been introduced by existing work, the main contribution of the paper is to show that the meta-learning of a "dataset-conditioned architecture generation" framework can enable fast generation of a good architecture without training on the target dataset. The proposed method is interesting and effective, however it requires an existing pool of good architectures for a given task, which may limit its applicability to a diverse set of real-world problems. I strongly encourage the authors to include experiments on a larger pool of architectures than the NAS-Bench-201 search space to show the strength of their proposed method in generating good architectures. While training MetaD2A with pairs of MetaImageNet and randomly sampled graph shows that the proposed framework can generate graphs with different types of edges, it doesn't show that it can successfully meta-learn to produce better architectures for a new task from an existing pool of good architectures.

We believe that many of the reviewers comments were addressed in the rebuttal, so while the scores are low, they do not reflect neither the contribution nor the reviewers opinion well (e.g., R3, in his last post, seems to suggest that his review should be updated but it has not happened).